# The Five-Year Prospective Study of Quality of Life in Hemifacial Spasm Treated with Abo-Botulinum Toxin A

**DOI:** 10.3390/toxins13030215

**Published:** 2021-03-16

**Authors:** Subsai Kongsaengdao, Narong Maneeton, Benchalak Maneeton

**Affiliations:** 1Division of Neurology, Department of Medicine, Rajavithi Hospital, Department of Medical Services, Public Health Ministry, Bangkok 10400, Thailand; 2Division of Neurology, Department of Medicine, Collage of Medicine, Rangsit University, Bangkok 10400, Thailand; 3Department of Psychiatry, Chiang Mai University, Chiang Mai 50200, Thailand; narong.m@cmu.ac.th (N.M.); benchalak.maneeton@cmu.ac.th (B.M.)

**Keywords:** hemifacial spasm, Abo-botulinum toxin A, the health-related quality of life, depressive disorder, 36-item short form health survey, AIMS

## Abstract

This study aimed to determine the long-term quality of life (QoL) in hemifacial spasm (HFS) patients after treating with Abo-botulinum toxin A (Abo-BTX). The study assessed the disease-specific QoL (hemifacial spasm questionnaire 30 items; HFS 30), the involuntary movements (abnormal involuntary movement scale; AIMS), general health QoL (Medical Outcomes 36-Item Short Form Health Survey; SF-36), and Depression (the Center of Epidemiologic Studies-Depression questionnaire; CES-D). A total of 74 HFS patients were enrolled from 2012 to 2017. The disease-specific QoL; involuntary movements; and the general health domain of SF 36 were significantly improved after injections of Abo-BTX A in the first few years (*p* < 0.04), but significantly decreased at the fifth year of treatment without significant clinical resistance observed (*p* < 0.001). Only the general health domain of SF 36 showed persistent improvement over five years (*p* = 0.02). In summary, Abo-BTX A can improved quality of life in the first few years; however only the general health domain of SF-36 showed significant improvement over five years (*p* = 0.02). No clinical resistance was observed.

## 1. Introduction

Botulinum toxin A is the gold standard of treatment in hemifacial spasm (HFS), the most common abnormal movement of the face, presented with brief, repetitive, involuntary tonic, and clonic contraction of facial expression muscle. Disability of HFS is directly from impaired vision and limit activity daily living. Global incidence for HFS is 0.8 per 100,000 persons, predominant in female patients (14.5 per 100,000) when compared to males (7.4 per 100,000) [1,2]. In Thailand, the incidence is approximately 9.8 per 100,000 persons [3].

Several botulinum toxins were approved for HFS treatment, Ona-botulinum toxin A (Ona-BTX A), Abo-botulinum toxin A (Abo-BTX A), Inca-botulinum toxin A and Rima-botulinum toxin B. Botulinum toxin A (BTX A) has the best efficacy in the treatment of HFS and improves the quality of life [4,5,6,7], especially Ona-BTX A [8] and Abo-BTX A [9]. Many studies including, a 10-year retrospective multicenter study proved, the efficacy of Ona-BTX A treatment in hemifacial spasm with minimal and transient adverse reaction [8], while a four-year prospective study confirmed the efficacy of Abo-BTX A in hemifacial spasm after 12 injection [9]. Only one short-term study in Thai patients reported that Abo-BTX A and Neu-botulinum toxin A (Neu-BTX A) improved only disease-specific quality of life (HFS 30 and AIMS) after 24-week treatment without any improvement of general quality of life (SF-36) [7]. Both efficacy of treatment and the health-related quality of life in HFS patients outcomes have been reported worldwide [4,5,6,7]. However, the study of health-related QoL has not been demonstrated with long-term outcomes after treatment with botulinum toxin A [8,9]. The HRQoL improves in HFS patients, possibly related to improvement of vision interference during reading and writing, social embarrassment, pain, depression, and anxiety, of which most publications showed depression is the comorbidity of hemifacial spasm.

The aim of present study was to determine the disease-specific and general HRQoL and depressive symptoms after twenty injections of Abo-BTX A over five years in Thai CD patients.

## 2. Results

In total, 56 females and 18 males, ranging in age from 20 to 75 years (mean ± standard deviation of 60.8 ± 7.5 years), were eligible in the study from February 2012 to February 2017. Those patients were included in the analysis. The mean ± standard deviation duration of symptoms of the patients was 5.27 ± 3.76 years. The mean ± standard deviation of previous Abo-BTX A injections were 10.85 ± 5.95 injection times.

### 2.1. The Disease-Specific HRQoL

After the twenty injections of Abo-BTX A every three month, the mean ± standard error of the HFS 30 significantly decreased from 30.1 ± 2.3 at the baseline to, at maximum reduction, 25.9 ± 2.1 at nine months and 29.0 ± 2.2 at 45 months. After 48 months, mean score ± standard error of the HFS 30 significantly increased to 39.2 + 4.4 (*p =* 0.004). There were no significant changes of mean score ± standard error AIMS in the first 15 months after treatment (*p =* 0.305). Interestingly, mean score + standard error of AIMS significantly decreased from 14.2 ± 1.0 at 18 months to, at maximum reduction, 11.2 ± 0.9 at 42 months (*p =* 0.041). After 45 months, mean score ± standard error of the HFS 30 significantly increased to 15.9 ± 2.3 (*p =* 0.008) (Figure 1 and Figure 2).

### 2.2. The general HRQoL

After twenty injections with Abo-BTX A treatment at three-month intervals, the mean ± standard error of the SF-36 significantly improved from 57.4 ± 2.4 at the baseline to, at maximum improvement, 61.3 ± 3.0 at 27 months. After 30 months, mean ± standard error of the SF-36 significantly declined to 52.1 ± 4.2 at 54 months (*p* = 0.002). Interestingly, the general health subdomain of SF-36 showed consistent improvement of average mean ± standard error of first year from 45.8 ± 1.5 to 50 ± 4.6 at the fifth year (*p* = 0.02) (Figure 1). For the physical functioning subdomain, the mean ± standard error significantly improved from the baseline (61.4 ± 3.6) to, at maximum reduction, 67.8 ± 4.2 at nine months, 65.7 ± 5.8 at 12 months, and 64.8 ± 4.2 at 15 months. After 18 months, mean ± standard error of the mean ± standard significantly reduced to 50 ± 6.2 at 57 months (*p* = 0.001). For the social functioning subdomain, bodily pain, mental health, and vitality showed no significant changes across the treatment (*p* = 0.569, 0.948, 0.137, and 0.322 respectively) (Figure 3 and Figure 4).

### 2.3. The CES-D Scores

Even though most of the patients seemed to be distressed from HFS, with cutoff scores of 20 or more for depressive disorder, there was no significant change from pre- to post-treatment based on the CES-D. According to the sequential analysis, no trend of the CES-D scores reduction was observed (*p* = 0.924) (Figure 1 and Figure 2). No patients were treated with anti-depressive medication.

## 3. Discussion

In 2016, the movement disorder group of the Cochrane systematic review reported only one small poorly designed double-blind, randomized, placebo-controlled study to prove the efficacy of botulinum toxin A in hemifacial spasm without impact of botulinum toxin A on QoL [10]. The previous and present study in Thai HFS patients showed benefit of short- and long-term treatment with Abo-BTX A, which could improve the disease-specific HRQoL, measured by the HFS 30 and AIMS scale [7]. This study showed peak of maximum improvement of HFS 30 at nine months and lasting for 45 months, as well as AIMS that showed peak of maximum improvement at 18 months and lasting for 42 months. Although the previous six-month treatment did not improve the general HRQoL [7], measured by the SF-36, this long-term study showed improvement lasting for 27 months after treatment, declining in the third and fourth years. Only the general health subdomain of SF-36 showed significant improvement after five years. The other subdomains of SF-36 were worsening in the fifth year of treatment without clinical efficacy decline. This worsening of general QoL occurred in the physical functioning subdomain, which may be due to the decrease of efficacy to improve physical functioning of HFS patients. This finding might be because botulinum toxin cannot improve all domains of SF-36 after the second year of treatment in HFS patients except general health domain, which showed more specific changes related to the botulinum toxin treatment across five years.

Previous evidence indicates that BTX A has shown its efficacy in treatment of HFS over 10-year [6] treatment; however, although the neutralized antibody in hemifacial spasm patients may be found in 1% of low protein content formulation Ona-BTX A, the neutralizing antibody Abo-BTX A has not been studied in hemifacial spasm patient. In this study, most likely reason there was a decrease in quality of life in the fifth year of treatment is the neutralizing antibody’s effect decreased to a sub-clinical response after long-term treatment. The association of neutralized antibody against Abo-BTX A and decrease of long-term QoL should be observed in further studies.

Although there is a lack of evidence in the aspect of quality of life available in hemifacial spasm comparing botulinum toxin with microvascular decompression, most safety and efficacy studies reported botulinum toxin has faster onset of improvement, is less invasive, and has lower complication rates when compared to MVD, which is more invasive, has longer onset of improvement, and has a higher rate of serious complications. For the quality-of-life aspect, most microvascular decompression studies report only short term, postoperative and one-year improvement of disease-specific QoL. The authors of [11,12] reported that 24.1% of re-operated microvascular decompression surgery patient have complications such as hearing loss and facial paresis, which have highly negative impacts on quality of life. Based on the updated evidence, this study reports long-term disease-specific QoL improvement without serious complications. Not surprisingly, botulinum toxin A is the only treatment proved as standard treatment with less serious complications, less invasiveness, fewer side effects, and improved long-term quality of life.

The mechanism of botulinum toxin A is a direct effect on the SNARE protein complex; synaptobrevin II results inhibit exocytosis of acetylcholine vesicles of facial motor neuron. Moreover, botulinum toxin A also indirectly inhibits glutamate release from cerebrocortical synaptosomes and modulates release of noradrenaline in PC12 cells by reduced signal input to central nervous system [13], suggesting a possible mechanism for a botulinum toxin A effect on depression and pain transmission in hemifacial spasm patients. In a previous study, injection of botulinum toxin A showed improvement of depressive symptoms in short-term treatment [7], but no significant changes of depressive symptoms in long-term treatment were found in this study. This may explain why the effect of Abo-BTX A, which reduced glutamate and increased noradrenaline in cerebrocortical pathway, is not long-lasting, since the depressive symptoms and bodily pain domain of SF-36 show clearly no significant change after five-year treatment. Interestingly, the bodily pain domain showed borderline significant (*p* = 0.055) reduction of pain score at second year of treatment. This finding may be explained by the adaptation of central nervous system by diminishing the effect of glutamate inhibition and increased noradrenaline release in cerebrocortical pathway in depression, after two-year treatment.

## 4. Conclusions

Abo-BTX A improved QoL in the first few years in HFS patients. Only general health domain showed significant improvement across five-year treatment. A decrease of QoL in the other domains was found in the fifth year after treatment without observing clinical resistance. Long-term treatment with botulinum toxin showed no effect on depressive symptoms in HFS patients. The association of neutralized antibody against Abo-BTX A and decrease of long-term QoL should be observed in further studies.

## 5. Materials and Methods

The five-year prospective study of the quality of life for HFS patients treated with Abo-BTX A injections at three-month intervals was designed following the Declaration of Helsinki and International Conference on Harmonization/Good Clinical Practice Guidelines. The research protocol was approved by the independent Ethics Committee. The informed consent of all patients was obtained before commencing all study procedures.

All HFS patients, aged between 18 and 80 years, who provided reliable answers to the HRQoL questionnaires (Thai version) were included in this study. The present study was approved (11/5/2011) and monitored by the Ethics Committee of Rajavithi Hospital. All participants gave written informed consent. (Study code 53263_082/2554). Patients who did not sign inform consent; were potentially pregnant without proper contraception or lactating; with significant medical conditions that could influence trial evaluations, for example bleeding tendency, significant cardiovascular, rheumatological, or neuropsychiatric disease except depressive disorder; or a history of allergy to BTX A were excluded.

### 5.1. Study Medication

First, 500 units of Abo-BTX A, freeze-dried powder, were diluted into 1.5 mL of normal saline solution, approximately 333.33 units of Abo-BTX A per 1.0 mL, and used within 2 h after dilution. Twenty injections, one every three months, to a total of 100 unit of Abo-BTX A, as intramuscular injections in ipsilateral orbicularis occuli and orbicularis oris (25 unit per injected site, for upper outer and lower outer of each muscle), was performed using 29G insulin needles for each patient (Figure 5).

### 5.2. Assessments

After performing a complete history and physical examination, all patients were assigned to complete the HFS 30, AIMS, Medical Outcomes 36-Item Short Form Health Survey (SF-36), and CES-D questionnaires at baseline and before the next visit. The primary outcome was the change from baseline in mean HFS 30, AIMS, and SF-36 questionnaire, after five years of injections of Abo-BTX A. The secondary endpoints were the changes in the mean CES-D score.

### 5.3. Efficacy Measurement

#### 5.3.1. The Disease-Specific HRQoL Questionnaire

The HFS 30, a validated questionnaire with good reliability of disease-specific HRQoL in patients with HFS, is composed of seven subscales (30 items): mobility (5 items), activity daily living (5 items), and communication (3 items) are classified as physical domain; emotional wellbeing (7 items), stigma (4 items), social support (3 items), and cognition (3 items), are classified as mental health domain. All the items are scored on a 5-point scale ranging from 0 (never) to 4 (always) with a consistency of Cronbach’s alpha >0.7, it was used to evaluate the HRQoL pre- and post-treatment of twenty injections of Abo-BTX A [8].

#### 5.3.2. Abnormal Involuntary Movement Scale (AIMS) (Thai Version)

The abnormal involuntary movement scale (AIMS) rates in seven regions the severity of face and mouth movements, each on a five-point scale in the overall severity, judged on the amplitude of muscle movements, incapacitation postures, and positions. Open mouth and dental status with absence of problems with teeth or dentures was also evaluated. The AIMS (Thai version) was subjected to test–retest to confirm the reliability coefficient with Cronbach’s alpha of >0.7 at Rajvithi Hospital [7].

#### 5.3.3. The General HRQoL Questionnaire

The SF-36 scale is composed of 36 items with eight domains: physical functioning (PF), role limitations due to physical health (RP), role limitations due to emotional problems (RE), vitality (VT), mental health (MH), social functioning (SF), bodily pain (BP), and general health (GH). The RE, VT, MH, and SF domains assess the mental aspect and PF, RP, and BP domains evaluate the physical health aspect of the HFS patients. The Thai version of SF-36 has been validated and tested for reliability in HFS patients [7,14,15,16].

#### 5.3.4. Depression Screening Questionnaire

CES-D questionnaire has 20 items with six subscales reflecting the major symptoms of depression: depressed mood, feelings of guilt, worthlessness and helplessness, psychomotor retardation, loss of appetite, and sleep difficulties. It is used to identify depressive symptoms for the general population in relation to major or clinical depression. The CES-D score ranges from 0 to 3 for each item (0 = rarely or none of the time, 1 = some or little of the time, 2 = moderately or much of the time, and 3 = most or almost all the time); the sum of its scores, therefore, ranges from 0 to 60, with higher scores demonstrating greater depressive symptoms. The scores of 20 or more determined risk of clinical depression with sensitivity of 79% and specificity of 80%. The Thai version CES-D has been validated and tested for reliability in Thai people [7,17].

### 5.4. Statistical Analysis

The mean scores (standard error) of QoL and CES-D were analyzed by general linear model by repeated measurement method, every three months of the five-year treatment. The tests between visits, considering linear, quadratic, and cubic significant models, were used to consider the maximum effect of mean change over the entire five-year treatment. The average one-year data were also analyzed by general linear model by repeated measurement method for five years, while the linear, quadratic, and cubic significant models were used to consider the maximum effect of mean change from baseline. All data were analyzed by using the Cytel^®^ Studio^®^ (license no 2060107) software package. All statistical tests were two-tailed with a significance level of α = 0.05 (Figure 5).

## Figures and Tables

**Figure 1 toxins-13-00215-f001:**
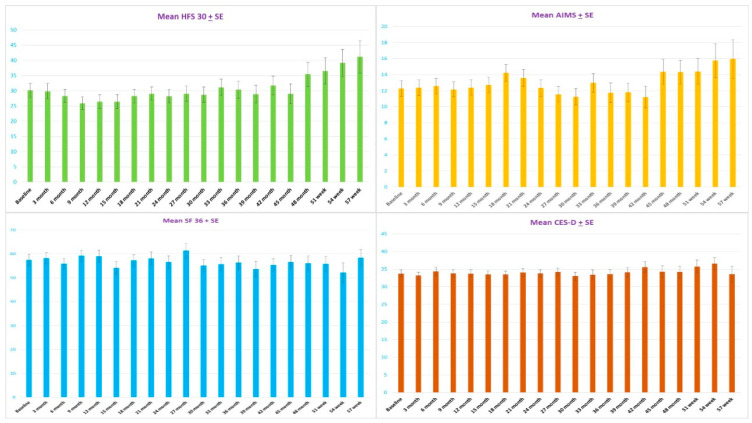
The mean ± standard error (SE) of disease specific quality of life (HF30) with significant linear model (*p =* 0.004) and quadratic model (*p =* 0.002) (**top left**); the abnormal involuntary movement scale (AIMS) with significant linear model (*p =* 0.041) at second year and significant linear model (*p =* 0.009) at fourth year (**top right**); general health quality of life (SF-36) with significant linear model (*p =* 0.031) at fourth year after injections of Abo-BTX A (**bottom left**); and the mean ± standard error (SE) of the Center of Epidemiologic Studies-Depression (CES-D) questionnaire with no significant improvement after injections of Abo-BTX A (*p =* 0.924) (**bottom right**).

**Figure 2 toxins-13-00215-f002:**
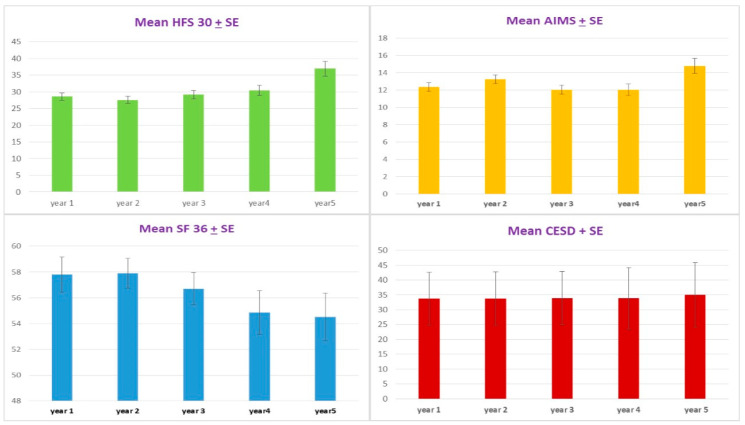
The mean ± standard error (SE) by year of disease-specific QoL (HF30) with significant linear model (*p =* 0.0001) (**top left**); the abnormal involuntary movement scale (AIMS) with significant linear model (*p =* 0.008) (**top right**); general health QoL (SF-36) showed significant linear model (*p =* 0.002) after injections of Abo-BTX A (**bottom left**); and the mean ± standard (SE) by year of Center of Epidemiologic Studies-Depression (CES-D) questionnaire showed no significant improvement after injections of Abo-BTX A (*p =* 0.191) (**bottom right**).

**Figure 3 toxins-13-00215-f003:**
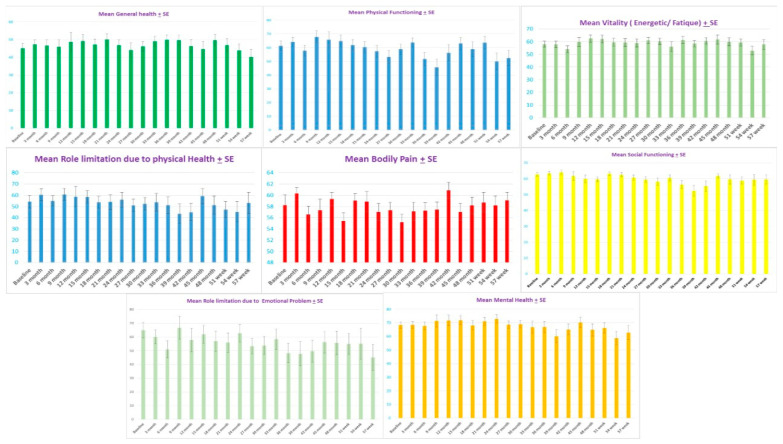
The mean ± standard error (SE) of physical functioning subdomain of SF-36 (**top middle**) showed significant improvement from baseline (61.4 ± 3.6) to the maximum reduction at nine months (67.8 ± 4.2) and lasting improvement at 12 months (65.7 ± 5.8) and 15 months (64.8 ± 4.2). The mean ± standard error (SE) of physical functioning subdomain of SF-36 after 18 months showed significantly reduced to 50 + 6.2 at 57 months (*p =* 0.001). The mean ± standard error (SE) of general health subdomain of SF-36 (**top left**) showed borderline significant improvement from baseline 45.1 ± 2.6 to maximum reduction at 36 months (49.8 ± 2.8) (linear model, *p =* 0.06) and, for social functioning subdomain (**middle right**), bodily pain (**middle**), mental health (**bottom right**), and vitality (**top right**) showed no significant changes across treatment (*p =* 0.569, 0.948, 0.137, and 0.322, respectively). The mean ± standard error (SE) of role limitation due to physical health subdomain of SF-36 (**middle left**) showed borderline significant decline from baseline 54.3 ± 5.5 to maximum declined at 54 months (45.2 ± 9.1) (quadratic model, *p =* 0.06 at fifth year), The mean ± standard error (SE) of role limitation due to emotional problems subdomain of SF-36 (**bottom left**) showed significant decline from baseline 64.9 ± 5.6 to, at maximum decline, 45.1 ± 9.3 at 57 months (quadratic model, *p =* 0.013).

**Figure 4 toxins-13-00215-f004:**
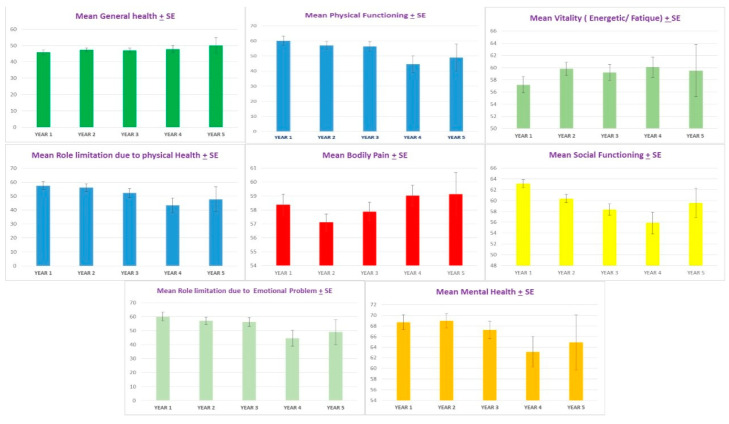
The mean ± standard error (SE) by year of general health subdomain of SF-36 (top left) showed significant improvement from baseline across five years (*p =* 0.02). The mean ± standard error (SE) by year of physical health subdomain of SF-36 (**top middle**), physical functioning subdomain of SF-36 (**top middle**), and emotional subdomain of SF-36 (**bottom left**) showed significant worsening at 4–5 years (*p =* 0.04, *p =* 0.03, and *p =* 0.03, respectively). The mean ± standard error (SE) by year of social functioning subdomain of SF-36 (**right middle**), bodily pain subdomain of SF-36 (**middle**), the mean + standard error (SE) health subdomain of SF-36 (**bottom right**), and vitality subdomain of SF-36 (**top right**) showed no significant changes across the five years of treatment (*p =* 0.56, 0.94, 1.00, and 0.56, respectively).

**Figure 5 toxins-13-00215-f005:**
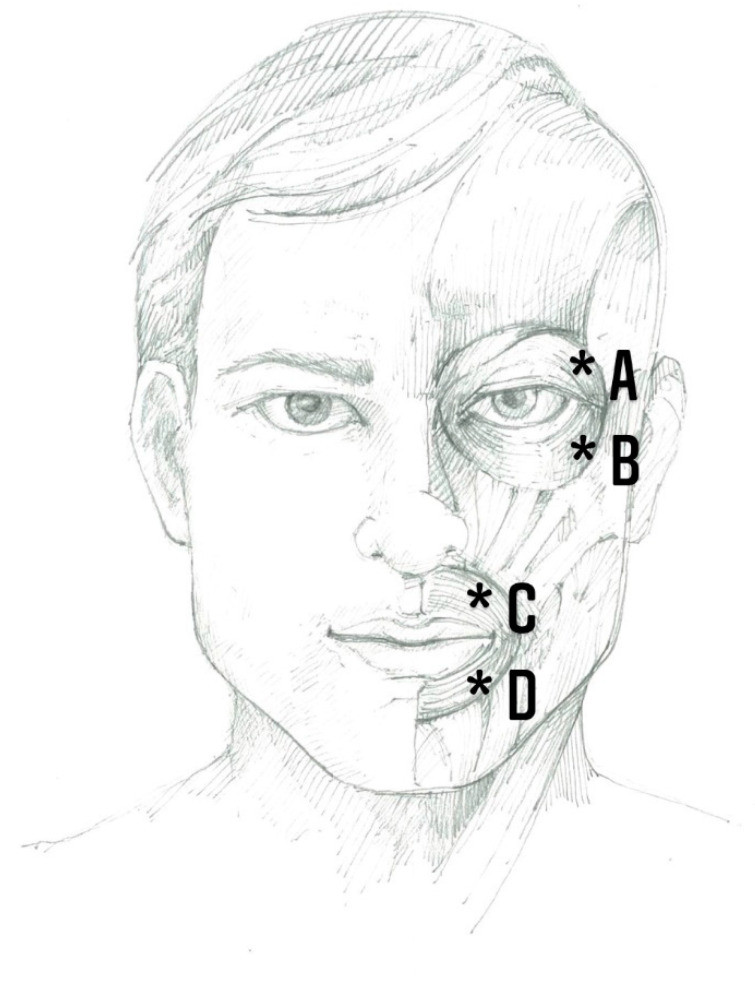
The intramuscular injection sites (*): upper outer border of orbicularis occuli (**A**); lower outer border of orbicularis occuli (**B**); upper outer border of orbicularis oris (**C**); and lower outer border of orbicularis oris (**D**). Twenty-five units of Abo-botulinum toxin A (Abo-BTX A) were injected per injected site.

## Data Availability

Not applicable.

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
