# Peer review of "The Five-Year Prospective Study of Quality of Life in Hemifacial Spasm Treated with Abo-Botulinum Toxin A"

_toxins, 2021, doi:10.3390/toxins13030215_

Round 1
Reviewer 1 Report
The results of treatment of HFS (and QoL !) significantly depends on the injection technique (unilaterally or bilaterally, injecting all target muscles or not, detailed doses, etc). You showed great statistics but poor clinical picture.
Additional conclusions can be:
1) to revise, change and improve the injection technique
2) to add antidepressants in the treatment
Author Response
The results of treatment of HFS (and QoL !) significantly depends on the injection technique (unilaterally or bilaterally, injecting all target muscles or not, detailed doses, etc). You showed great statistics but poor clinical picture.
Answer: the clinical picture illustration was added as figure 1
Additional conclusions can be:
- to revise, change and improve the injection technique
Answer: details were added by the line 74-78
- to add antidepressants in the treatment
Answer: details were added by the line 74-78
Reviewer 2 Report
INTRODUCTION - there is lapsus in last sentence that authors have follow patients during 2 years instead 5 years period
MATERIALS AND METHODS - authors should stated when they performed second testings after the BonT injection. Scales are not described properly - data about evaluation of scores is necessary to understand the results. Beside this, HFS30 is described unclearly. Why use AIMS instead Jankovic Rating Scale of HF severity scale? Did you test normality of data distribution and what test used?
RESULTS - Figures do not have title. Under the figures is unnecessary written results instead in text.
DISCUSSION - not clearly written. How did you conclude that BonT influence on depression with your results?? How did you conclude from your results that QoL decreased due to neutralizing antibodies?
English language should be corrected.
Author Response
INTRODUCTION - there is lapsus in last sentence that authors have follow patients during 2 years instead 5 years period
Answer: Changed to 5 years line 62
MATERIALS AND METHODS - authors should stated when they performed second testings after the BonT injection
Answer: details added in line 81-82
. Scales are not described properly - data about evaluation of scores is necessary to understand the results
Answer: scale are describe as details in line 85-111
. Beside this, HFS30 is described unclearly.
Answer: it is standard detail of this test
Why use AIMS instead Jankovic Rating Scale of HF severity scale?
Answer: this AIMS had already validated and published in Thai version and we used as reference to previous study compare
Did you test normality of data distribution and what test used?
Answer: the multiple repeated analysis had been tested the normality of data by Cytel studio progeram
RESULTS - Figures do not have title. Under the figures is unnecessary written results instead in text.
Answer: confirm no change
DISCUSSION - not clearly written. How did you conclude that BonT influence on depression with your results??
Answer: the reference 7 showed short term study had effect to depression but not for long term in this study
How did you conclude from your results that QoL decreased due to neutralizing antibodies?
Answer: just the discussion for possible mechanism
English language should be corrected.
Round 2
Reviewer 2 Report
Manuscript was corrected properly.